# Spectral Graph Wavelets for Structural Role Similarity in Networks

## Abstract

Nodes residing in different parts of a graph can have similar structural roles within their local network topology. The identification of such roles provides key insight into the organization of networks and can also be used to inform machine learning on graphs. However, learning structural representations of nodes is a challenging unsupervised-learning task, which typically involves manually specifying and tailoring topological features for each node. Here we develop GraphWave, a method that represents each node's local network neighborhood via a low-dimensional embedding by leveraging spectral graph wavelet diffusion patterns. We prove that nodes with similar local network neighborhoods will have similar GraphWave embeddings even though these nodes may reside in very different parts of the network. Our method scales linearly with the number of edges and does not require any hand-tailoring of topological features. We evaluate performance on both synthetic and real-world datasets, obtaining improvements of up to 71% over state-of-the-art baselines.

## 1 Introduction

Structural role discovery in graphs focuses on identifying nodes which have topologically similar local neighborhoods (*i.e.*, similar local structural roles) while residing in potentially distant areas of the network (Figure 1). Such alternative definition of node similarity is very different than more traditional notions (Perozzi et al., 2014; Grover and Leskovec, 2016; Yang et al., 2016; Monti et al., 2016; Kipf and Welling, 2017; Hamilton et al., 2017a;b; Garcia-Duran and Niepert, 2017), which all assume some notion of "smoothness" over the graph and thus consider nodes residing in close network proximity to be similar. Such structural role information about the nodes can be used for a variety of tasks, including as input to machine learning problems, or even to identify key nodes in a system (principal "influencers" in a social network, critical hubs in contagion graphs, etc.).

When structural roles of nodes are defined over a discrete space, they correspond to different topologies of local network neighborhoods (*e.g.*, edge of a chain, center of a star, a bridge between two clusters). However, such discrete roles must be pre-defined, requiring domain expertise and manual inspection of the graph structure. A more powerful and robust method for identifying structural similarity involves learning a continuous vector-valued *structural signature* $\chi_a$ of each node $a$ in an unsupervised way. This motivates a natural definition of structural similarity in terms of closeness of topological signatures: For any $\epsilon > 0$, nodes $a$ and $b$ are defined to be $\epsilon$-structurally similar with respect to a given distance if: $dist(\chi_a, \chi_b) \leq \epsilon$. Thus, a robust structural similarity metric must introduce both an appropriate signature and an adequate distance metric.

While several methods have been proposed for structural role discovery in graphs, existing approaches are extremely sensitive to small perturbations in the topology and typically lack one or more desirable properties. They often require manually hand-labeling topological features (Henderson et al., 2012), rely on non-scalable heuristics (Ribeiro et al., 2017), and/or return a single similarity score instead of a multidimensional structural signature (Jin et al., 2011; 2014).

Here we address the problem of structure learning on graphs by developing GraphWave. Building upon techniques from graph signal processing (Coifman et al., 2006; Hammond et al., 2011; Shuman et al., 2013), our approach learns a structural embedding for each node based on the diffusion of a spectral graph wavelet centered at that node. Intuitively, each node propagates a unit of energy over the graph and characterizes its neighboring topology based on the response of the network to

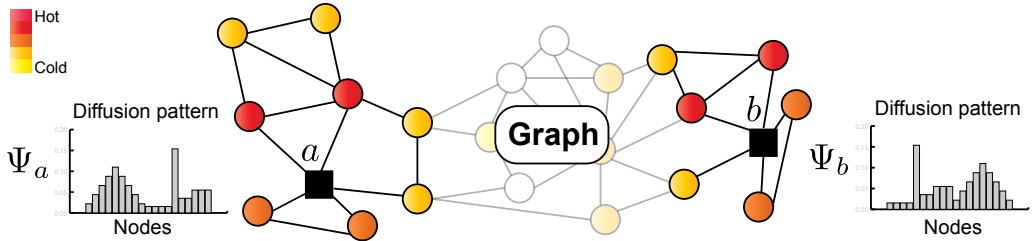

Figure 1: Nodes $a$ and $b$ have similar local structural roles even though they are distant in the graph. While raw spectral graph wavelet signatures/coefficients $\Psi$ of $a$ and $b$ might be very different, we treat them as probability distributions and show that the coefficient distributions are indeed similar.

this probe. In contrast to prior work that characterizes the wavelet diffusion as a function of the wavelet scaling parameter, we study how the wavelet diffuses through the network at a given scale as a function of the initial source node. We prove that the coefficients of this wavelet directly relate to graph topological properties. Hence, these coefficients contain all the necessary information to recover structurally similar nodes, without requiring the hand-labeling of features. However, the wavelets are, by design, localized on the graph. Therefore to compare structural signatures for nodes that are far away from each other, typical graph signal processing methods (using metrics like correlation between wavelets or $\ell_2$ distance) cannot be used without specifying an exact one-to-one mapping between nodes for every pairwise comparison, a computationally intractable task.

To overcome this challenge, we propose a novel way of treating the wavelets as probability distributions over the graph. This way the structural information is contained in *how* the diffusion spreads over the network rather than *where* it spreads. In order to provide vector-valued signatures which can then be used as input to any machine learning algorithm, we embed these wavelet distributions using the empirical characteristic function (Lukacs, 1970). The advantage of empirical characteristic functions is that they capture all the moments of a given distribution. This allows GRAPHWAVE to be robust to small perturbations in the local edge structure, as we prove mathematically. Computational complexity of GRAPHWAVE is linear in the number of edges, thus allowing it to scale to large (sparse) networks. Finally, we compare GRAPHWAVE to several state-of-the-art baselines on both real and synthetic datasets, obtaining improvements of up to 71% and demonstrating how our approach is a useful tool for characterizing structural signatures in graphs.

**Summary of contributions.** The main contributions of our paper are as follows:

- We develop a novel use of spectral graph wavelets by treating them as probability distributions and characterizing the distributions using empirical characteristic functions.
- We leverage these insights to develop a scalable method (GRAPHWAVE) for learning node embeddings based on structural similarity in graphs.
- We prove that GRAPHWAVE accurately recovers structurally similar nodes.

**Further related work.** Prior work on discovering nodes with similar structural roles has typically relied on explicit featurization of nodes. These methods generate an exhaustive listing of each node's local topological properties (*e.g.*, node degree, number of triangles it participates in, number of $k$-cliques, its PageRank score) before computing node similarities based on such heuristic representations. A notable example of such approaches is *RolX* (Henderson et al., 2012), which aims to recover a soft-clustering of nodes into a predetermined number of $K$ distinct roles using recursive feature extraction (Henderson et al., 2011). Similarly, *struc2vec* (Ribeiro et al., 2017) uses a heuristic to construct a multilayered graph based on topological metrics and simulates random walks on the graph to capture structural information. In contrast, our approach does not rely on heuristics (we mathematically prove its efficacy) and does not require explicit manual feature engineering or hand-tuning of parameters.

Another line of related work are graph diffusion kernels (Coifman et al., 2006) which have been utilized for various graph modeling purposes (Kondor and Lafferty, 2002; Chung, 2007; Rustamov and Guibas, 2013; Tremblay et al., 2014). However, to the best of our knowledge, our paper is the first to apply graph diffusion kernels for determining structural roles in graphs. Kernels have been shown to efficiently capture geometrical properties and have been successfully used for shape detection in the image processing community (Sun et al., 2009; Ovsjanikov et al., 2010; Aubry et al.,

2011). However, in contrast to shape-matching problems, GRAPHWAVE considers these kernels as probability distributions over real-world graphs. This is because the graphs that we consider are highly irregular (as opposed to the Euclidean and manifold graphs). Therefore, traditional wavelet methods, which typically analyze node diffusions across specific nodes that occur in regular and predictable patterns, do not apply. Instead, by treating wavelets as distributions, GRAPHWAVE characterizes the *shape* of the diffusion, rather than the specific nodes where the diffusion occurs. This key insight allows us to uncover structural signatures and to discover structurally similar nodes.

## 2    LEARNING STRUCTURAL SIGNATURES

Given an undirected connected graph $\mathcal{G} = (\mathcal{V}, \mathcal{E})$ with $N$ nodes $\mathcal{V} = \{a_1, \ldots, a_N\}$, edges $\mathcal{E}$, an adjacency matrix $A$ (binary or weighted), and a degree matrix $D_{ii} = \sum_j A_{ij}$, we consider the problem of learning, for every node $a_i$, a *structural signature* representing $a_i$'s position in a continuous multidimensional space of structural roles.

We frame this as an unsupervised learning problem based on spectral graph wavelets (Hammond et al., 2011) and develop an approach called GRAPHWAVE that provides mathematical guarantees on the optimality of learned structural signatures.

### 2.1    SPECTRAL GRAPH WAVELETS

In this section, we provide background on a spectral graph wavelet-based model (Hammond et al., 2011; Shuman et al., 2013) that we will use in the rest of the paper.

Let $U$ be the eigenvector decomposition of the unnormalized graph Laplacian $L = D - A$ and let $\lambda_1 < \lambda_2 \leq \cdots \leq \lambda_N$ ($\Lambda = \text{Diag}(\lambda_1, \ldots, \lambda_N)$) denote the eigenvalues of $L$. Let $g_s$ be a filter kernel with scaling parameter $s$. For simplicity, we use the heat kernel $g_s(\lambda) = e^{-\lambda s}$ throughout this paper, but our results apply to any low-pass filter kernel (Shuman et al., 2016). For now, we assume that $s$ is given; we develop a method for selecting an appropriate value of $s$ in Appendix C.

Graph signal processing (Hammond et al., 2011; Shuman et al., 2013) defines the spectral graph wavelet associated with $g_s$ as the signal resulting from the modulation in the spectral domain of a Dirac signal centered around node $a$. The spectral graph wavelet $\Psi_a$ is given by an $N$-dimensional vector:

$$\Psi_a = U \, \text{Diag}(g_s(\lambda_1), \ldots, g_s(\lambda_N)) U^T \delta_a, \tag{1}$$

where $\delta_a = \mathbb{1}(a)$ is the one-hot vector for node $a$. For notational simplicity, we drop the explicit dependency of spectral graph wavelet $\Psi_a$ on $s$. The $m$-th wavelet coefficient of this column vector is thus given by $\Psi_{ma} = \sum_{l=1}^{N} g_s(\lambda_l) U_{ml} U_{al}$.

In spectral graph wavelets, the kernel $g_s$ modulates the eigenspectrum such that the resulting signal is typically *localized on the graph and in the spectral domain* (Shuman et al., 2013). Spectral graph wavelets are based on an analogy between temporal frequencies of a signal and the Laplacian's eigenvalues. Eigenvectors associated with smaller eigenvalues carry slow varying signal, encouraging nodes that are geographically close in the graph to share similar values. In contrast, eigenvectors associated with larger eigenvalues carry faster-varying signal across edges. The low-pass filter kernel $g_s$ can thus be seen as a modulation operator that discounts higher eigenvalues and enforces smoothness in the signal variation on the graph.

### 2.2    GRAPHWAVE ALGORITHM

First we describe the GRAPHWAVE algorithm (Alg. 1) and then analyze it in the next section. For every node $a$, GRAPHWAVE returns a $2d$-dimensional vector $\chi_a$ representing its *structural signature*, where nodes with structurally similar local network neighborhoods will have similar signatures.

We first apply spectral graph wavelets to obtain a diffusion pattern for every node (Line 3), which we gather in a matrix $\Psi$. Here, $\Psi$ is a $N \times N$ matrix, where $a$-th column vector is the spectral graph wavelet for a heat kernel centered at node $a$. In contrast to prior work that studies wavelet coefficients as a function of the scaling parameter $s$, we study them as a function of the network (*i.e.*, how the coefficients vary across the local network neighborhood around the node $a$). In particular, coefficients in each wavelet are identified with the nodes and $\Psi_{ma}$ represents the amount of energy that node $a$

---

**Algorithm 1** GRAPHWAVE algorithm for learning structural signatures.

1: **Input:** Graph $\mathcal{G} = (\mathcal{V}, \mathcal{E})$, scale $s$, evenly spaced sampling points $\{t_1, t_2, \ldots, t_d\}$.
2: **Output:** Structural signature $\chi_a \in \mathbb{R}^{2d}$ for every node $a \in \mathcal{V}$
3: Compute $\Psi = U g_s(\Lambda) U^T$ (Eq. (1))
4: **for** $t \in \{t_1, t_2, \ldots, t_d\}$ **do**
5:     Compute $\phi(t) = $ column-wise mean$(e^{it\Psi}) \in \mathbb{R}^N$
6:     **for** $a \in \mathcal{V}$ **do**
7:         Append $\text{Re}(\phi_a(t))$ and $\text{Im}(\phi_a(t))$ to $\chi_a$

---

has received from node $m$. As we will later show nodes $a$ and $b$ with similar network neighborhoods have similar spectral wavelet coefficients $\Psi$ (assuming that we know how to solve the "isomorphism" problem and find the explicit one-to-one mapping of the nodes from $a$'s neighborhood to the nodes of the $b$'s neighborhood). To resolve the node mapping problem GRAPHWAVE treats the wavelet coefficients as a probability distribution and characterizes the distribution via empirical characteristic functions. This is the key insight that makes it possible for GRAPHWAVE to learn nodes' structural signatures via spectral graph wavelets.

More precisely, we embed spectral graph wavelet coefficient distributions into $2d$-dimensional space (Line 4-7) by calculating the characteristic function for each node's coefficients $\Psi_a$ and sample it at $d$ evenly spaced points. The characteristic function of a probability distribution $X$ is defined as: $\phi_X(t) = \mathbb{E}[e^{itX}], t \in \mathbb{R}$ (Lukacs, 1970). The function $\phi_X(t)$ fully characterizes the distribution of $X$ because it captures information about all the moments of probability distribution $X$ (Lukacs, 1970). For a given node $a$ and scale $s$, the empirical characteristic function of $\Psi_a$ is defined as:

$$\phi_a(t) = \frac{1}{N} \sum_{m=1}^{N} e^{it\Psi_{ma}} \tag{2}$$

Finally, structural signature $\chi_a$ of node $a$ is obtained by sampling the 2-dimensional parametric function (Eq. (2)) at $d$ evenly spaced points $t_1, \ldots t_d$ and concatenating the values:

$$\chi_a = \big[\text{Re}(\phi_a(t_i)), \text{Im}(\phi_a(t_i))\big]_{t_1, \cdots t_d} \tag{3}$$

Note that we sample/evaluate the empirical characteristic function $\phi_a(t)$ at $d$ points and this creates a structural signature of size $2d$. This means that the dimensionality of the structural signature is independent of the graph size. Furthermore, nodes from different graphs can be embedded into the same space and their structural roles can be compared across different graphs.

**Distance between structural signatures.** The final output of GRAPHWAVE is a structural signature $\chi_a$ for each node $a$ in the graph. We can explore distances between the signatures through the use of the $\ell_2$ distance on $\chi_a$. The structural distance between nodes $a$ and $b$ is then defined as: $dist(a, b) = \|\chi_a - \chi_b\|_2$. By definition of the characteristic function, this technique amounts to comparing moments of different orders defined on wavelet coefficient distributions.

**Scaling parameter.** The scaling parameter $s$ determines the radius of network neighborhood around each node $a$ (Tremblay et al. (2014); Hammond et al. (2011)). A small value of $s$ determines node signatures based on similarity of nodes' immediate neighborhoods. In contrast, a larger value of $s$ allows the diffusion process to spread farther in the network, resulting in signatures based on neighborhoods with greater radii.

GRAPHWAVE can also integrate information across different radii of neighborhoods by jointly considering many different values of $s$. This is achieved by concatenating $J$ representations $\chi_a^{(s_j)}$, each associated with a scale $s_j$, where $s_j \in [s_{\min}, s_{\max}]$. We provide a theoretically justified method for finding an appropriate range $s_{\min}$ and $s_{\max}$ in Appendix C. In this multiscale version of GRAPHWAVE, the final aggregated structural signature for node $a$ is a vector $\chi_a \in \mathbb{R}^{2dJ}$ with the following form: $\chi_a = [\text{Re}(\phi_a^{(s_j)}(t_i)), \text{Im}(\phi_a^{(s_j)}(t_i)]_{t_i, s_j}$.

**Computational complexity.** We use Chebyshev polynomials (Shuman et al., 2011) to compute Line 3 in Algorithm 1. As in Defferrard et al. (2016), each power of the Laplacian has a computational cost of $O(|\mathcal{E}|)$, yielding an overall complexity of $O(K|\mathcal{E}|)$, where $K$ denotes the order Chebyshev polynomial approximation. The overall complexity of GRAPHWAVE is linear in the number of edges, which allows GRAPHWAVE to scale to large sparse networks.

## 3 ANALYSIS OF GRAPHWAVE

In this section, we provide theoretical motivation for our spectral graph wavelet-based model (Shuman et al., 2013). First we analytically show that spectral graph wavelet coefficients characterize the topological structure of local network neighborhoods (Section 3.1). Then we show that structurally equivalent/similar nodes have near-identical/similar signatures (Sections 3.2 and 3.3), thereby providing a mathematical guarantee on the optimality of GRAPHWAVE.

### 3.1 SPECTRAL GRAPH WAVELETS AS A MEASURE OF NETWORK STRUCTURE

We start by establishing the relationship between the spectral graph wavelet of a given node $a$ and the topological properties of local network neighborhood centered at $a$. In particular, we prove that a wavelet coefficient $\Psi_{ma}$ provides a measure of network connectivity between nodes $a$ and $m$.

We use the fact that the spectrum of the graph Laplacian is discrete and contained in the compact set $[0, \lambda_N]$. It then follows from the Stone-Weierstrass theorem that the restriction of kernel $g_s$ to the interval $[0, \lambda_N]$ can be approximated by a polynomial. This polynomial approximation, denoted as $P$, is tight and its error can be uniformly bounded. Formally, this means:

$$\forall \epsilon > 0, \quad \exists P : P(\lambda) = \sum_{k=0}^{K} \alpha_k \lambda^k \text{ such that } \quad |g_s(\lambda) - P(\lambda)| \leq \epsilon \quad \forall \lambda \in [0, \lambda_{\max}], \quad (4)$$

where $K$ is the order of polynomial approximation, $\alpha_k$ are coefficients of the polynomial, and $r(\lambda) = g_s(\lambda) - P(\lambda)$ is the residual. We can now express the spectral graph wavelet for node $a$ in terms of the polynomial approximation as:

$$\Psi_a = (\sum_{k=0}^{K} \alpha_k L^k) \delta_a + U r(\Lambda) U^T \delta_a. \quad (5)$$

We note that $\Psi_a$ is a function of $L^k = (D - A)^k$ and thus can be interpreted using graph theory. In particular, it contains terms of the form $D^k$ (capturing the degree), $A^k$ (capturing the number of $k$-length paths that node $a$ participates in), and terms containing both $A$ and $D$, which denote paths of length up to $k$ going from node $a$ to every other node $m$.

Using the Cauchy-Schwartz's inequality and the facts that $U$ is unitary and $r(\lambda)$ is uniformly bounded (Eq. (4)), we can bound the second term on the right-hand side of Eq. (5) by:

$$|\delta_m^T U r(\Lambda) U^T \delta_a|^2 = |\sum_{j=1}^{N} r(\lambda_j) U_{aj} U_{mj}|^2 \leq \left(\sum_{j=1}^{N} |r(\lambda_j)|^2 U_{aj}^2\right)\left(\sum_{j=1}^{N} U_{mj}^2\right) \leq \epsilon^2. \quad (6)$$

As a consequence, each wavelet $\Psi_a$ can be approximated by a $K$-th order polynomial that captures information about the $K$-hop neighborhood of node $a$. The analysis of Eq. (5), where we show that the second term is limited by $\epsilon$, indicates that spectral graph wavelets are predominately governed by topological features (specifically, degrees, cycles and paths) according to the specified heat kernel. The wavelets thus contain the information necessary to generate structural signatures of nodes.

### 3.2 SIGNATURES OF STRUCTURALLY EQUIVALENT NODES

Let us consider nodes $a$ and $b$ whose $K$-hop neighborhoods are identical (where $K$ is an integer less than the diameter of the graph), meaning that nodes $a$ and $b$ are structurally equivalent. We now show that $a$ and $b$ have $\epsilon$-structurally similar signatures in GRAPHWAVE.

First, we use the Taylor expansion to obtain an explicit $K$-th order polynomial approximation of $g_s$ as: $P(\lambda, s) = \sum_{k=0}^{K} (-1)^k (s\lambda)^k / k!$. Then, for each eigenvalue $\lambda$, we use the Taylor-Lagrange equality to ensure the existence of $c_\lambda \in [0, s]$ such that:

$$|r(\lambda)| = |e^{-\lambda s} - P(\lambda, s)| = \frac{(\lambda s)^{K+1}}{(K+1)!} e^{-\lambda c_\lambda} \leq \frac{(\lambda s)^{K+1}}{(K+1)!}. \quad (7)$$

If we take any $s$ such that it satisfies: $s \leq ((K+1)!\epsilon))^{1/(K+1)}/\lambda_2$, then the absolute residual $|r(\lambda)|$ in Eq. (7) can be bounded by $\epsilon$ for each eigenvalue $\lambda$. Here, $\epsilon$ is a parameter that we can specify depending on how close we want the signatures of structurally equivalent nodes to be (note that smaller values of the scale $s$ lead to smaller values of $\epsilon$ and thus tighter bounds).

Because $a$ and $b$ are structurally equivalent, there exists a one-to-one mapping $\pi$ from the $K$-hop neighborhood of $a$ (*i.e.*, $\mathcal{N}_K(a)$) to the $K$-hop neighborhood of $b$ (*i.e.*, $\mathcal{N}_K(b)$), such that: $\mathcal{N}_K(b) = \pi(\mathcal{N}_K(a))$. We extend the mapping $\pi$ to the whole graph $\mathcal{G}$ by randomly mapping the remaining nodes. Following Eq. (5), we write the difference between each pair of mapped coefficients $\Psi_{ma}$ and $\Psi_{\pi(m)b}$ in terms of the $K$-th order approximation of the graph Laplacian:

$$|\Psi_{ma} - \Psi_{\pi(m)b}| = \left|\delta_m U(P(\Lambda) + r(\Lambda))U^T\delta_a - \delta_{\pi(m)}U(P(\Lambda) + r(\Lambda))U^T\delta_b\right|$$
$$\leq \left|(UP(\Lambda)U^T)_{ma} - (UP(\Lambda)U^T)_{\pi(m)a}\right| + \left|(Ur(\Lambda)U^T)_{ma}\right| + \left|(Ur(\Lambda)U^T)_{\pi(m)b}\right|. \quad (8)$$

Here, we analyze the first term on the second line in Eq. (8). Since the $K$-hop neighborhoods around $a$ and $b$ are identical and by the localization properties of the $k$-th power of the Laplacian ($k$-length paths, Section 3.1), the following holds:

$$\forall m \in \mathcal{N}_K(a), (\sum_{k=0}^{K} \alpha_k L^k)_{ma} = (\sum_{k=0}^{K} \alpha_k L^k)_{\pi(m)b},$$

$$\forall m \notin \mathcal{N}_K(a), (\sum_{k=0}^{K} \alpha_k L^k)_{ma} = (\sum_{k=0}^{K} \alpha_k L^k)_{\pi(m)a} = 0,$$

meaning that this term cancels out in Eq. (8). To analyze the second and third terms on the second line of Eq. (8), we use bound for the residual term in the spectral graph wavelet (Eq. (6)) to uniformly bound entries in matrix $Ur(\Lambda)U^T$ by $\epsilon$. Therefore, each wavelet coefficient in $\Psi_a$ is within $2\epsilon$ of its corresponding wavelet coefficient in $\Psi_b$, *i.e.*, $|\Psi_{ma} - \Psi_{\pi(m)b}| \leq 2\epsilon$. As a result, because similarity in distributions translates to similarity in the resulting characteristic functions (Lévy's continuity theorem), then assuming the appropriate selection of scale, structurally equivalent nodes have $\epsilon$-structurally similar signatures.

### 3.3 SIGNATURES OF STRUCTURALLY SIMILAR NODES

We now analyze structurally similar nodes, or nodes whose $K$-hop neighborhoods are identical up to a small perturbation of the edges. We show that such nodes have similar GRAPHWAVE signatures.

Let $\widetilde{\mathcal{N}}_K(a)$ denote a perturbed $K$-hop neighborhood of node $a$ obtained by rewiring edges in the original $K$-hop neighborhood $\mathcal{N}_K(a)$. We denote by $\tilde{L}$ the graph Laplacian associated with that perturbation. We next show that when perturbation of a node neighborhood is small, the changes in the wavelet coefficients for that node are small as well.

Formally, assuming a small perturbation of the graph structure (*i.e.*, $\sup ||L^k - \tilde{L}^k||_F \leq \epsilon$, for all $k \leq K$), we use $K$-th order Taylor expansion of kernel $g_s$ to express the wavelet coefficients in the perturbed graph as:

$$\widetilde{\Psi}_a = \sum_{k=0}^{K} \alpha_k \tilde{L}^k + \tilde{U}r(\tilde{\Lambda})\tilde{U}^T. \quad (9)$$

We then use the Weyl's theorem (Coburn et al., 1966) to relate perturbations in the graph structure to the change in the eigenvalues of the graph Laplacian. In particular, a small perturbation of the graph yields small perturbations of the eigenvalues. That is, for each $\tilde{\lambda}$, $r(\tilde{\lambda})$ is close its original value $r(\lambda)$: $r(\tilde{\lambda}) = r(\lambda) + o(\epsilon) \leq C\epsilon$, where $C$ is a constant. Taking everything together, we get:

$$|\Psi_{ma} - \widetilde{\Psi}_{ma}| \leq |\sum_{k=0}^{K} \alpha_k(L^k - \tilde{L}^k)_{ma}| + |\tilde{U}r(\tilde{\Lambda})\tilde{U}^T|_{ma} + |Ur(\Lambda)U^T|_{ma} = (\sum_{k=0}^{K} |\alpha_k| + 1 + C)\epsilon,$$

indicating that structurally similar nodes have similar signatures in GRAPHWAVE.

## 4 EXPERIMENTS ON SYNTHETIC GRAPHS

**Baselines.** We compare our GRAPHWAVE method against two state-of-the-art baselines, *struc2vec* (Ribeiro et al., 2017) and *RolX* (Henderson et al., 2012). We note that *RolX* requires the number of desired structural classes as input, whereas the two other methods learn embeddings that capture a continuous spectrum of roles rather than discrete classes. We thus use *RolX* as an oracle estimator, providing it with the correct number of classes[1]. We also note that homophily-based methods (Kipf and Welling (2017); Hamilton et al. (2017a), etc.) are unable to recover structural similarities.

---

[1] All code and data for experiments can be downloaded at: `https://goo.gl/ndBkwb`.

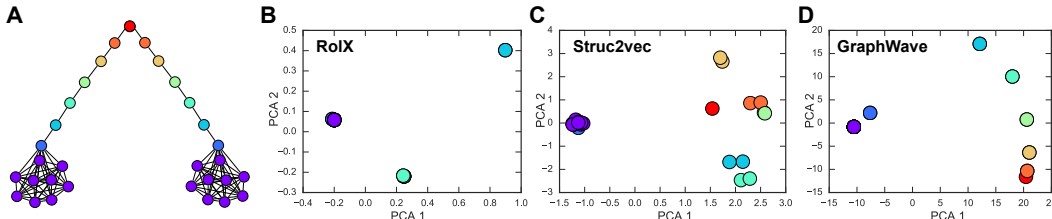

Figure 2: Barbell graph. The graph has 8 distinct classes of structurally equivalent nodes as indicated by color (**A**). 2D PCA projection of structural signatures as learned by *RolX* (**B**), *struc2vec* (**C**) and GRAPHWAVE (**D**). Projections in **B-D** contain the same number of points as there are nodes in the barbell graph. Identical signatures have identical projections, resulting in overlapping points in **B-D**.

## 4.1 BARBELL GRAPH

We consider a barbell graph consisting of two dense cliques connected by a long chain (Figure 2A). We run GRAPHWAVE, *RolX*, and *struc2vec* and plot a 2D PCA representation of learned structural signatures in Figure 2B-D.

GRAPHWAVE correctly learns identical representations for structurally equivalent nodes, providing empirical evidence for our theoretical result in Section 3.2. This can be seen by structurally equivalent nodes in Figure 2A (nodes of the same color) having identical projections in the PCA plot (Figure 2D). In contrast, both *RolX* and *struc2vec* fail to recover the exact structural equivalences.

All three methods correctly group the clique nodes (purple) together. However, only GRAPHWAVE correctly differentiates between nodes connecting the two dense cliques in the barbell graph, providing empirical evidence for our theoretical result in Section 3.3. GRAPHWAVE represents those nodes in a gradient-like pattern that captures the spectrum of structural roles of those nodes (Figure 2D).

## 4.2 COMPLEX AND NOISY GRAPH STRUCTURES

**Graphs.** We next consider four types of synthetic graphs where the structural role of each node is known and used as ground truth information to evaluate performance. The graphs are given by basic shapes of one of different types ("house", "fan", "star") that are regularly placed along a cycle (Table 1 and Figure 3A). In the "varied" setup, we mix the three basic shapes when placing them along a cycle, thus generating synthetic graphs with richer and more complex structural role patterns. Additional graphs are generated by placing these shapes irregularly along the cycle followed by adding a number of edges uniformly at random. In our experiments, we set this number to be around 5% of the edges in the original structure. This setup is designed to assess the robustness of the methods to data perturbations ("house perturbed", "varied perturbed").

**Experimental setup.** For each graph, we run *RolX*, *struc2vec*, and GRAPHWAVE to learn the signatures. We choose to use a multiscale version of GRAPHWAVE where the scale was set as explained in Appendix C. We then use $k$-means to cluster the learned signatures and use three standard metrics to evaluate the clustering quality. (1) *Cluster homogeneity* is the conditional entropy of the ground-truth structural roles given the proposed clustering (Rosenberg and Hirschberg, 2007). (2) *Cluster completeness* (Rosenberg and Hirschberg, 2007) evaluates whether nodes with the same structural role are in the same cluster. (3) *Silhouette score* compares the mean intra-cluster distance to the mean between-cluster distance, assessing the density of the recovered clusters. This score takes a value in [-1,1] (higher is better).

**Results.** GRAPHWAVE consistently outperforms *struc2vec*, yielding improvements for the homogeneity of up to 50%, and completeness up to 69% in the "varied" setting (Table 1). Both GRAPHWAVE and *RolX* achieved perfect performance in the noise-free "house" setting, however, GRAPHWAVE outperformed *RolX* by up to 4% (completeness) in the more complex "varied" setting. We evaluated methods on graphs in the presence of noise ("perturbed" in Table 1): GRAPHWAVE outperformed *RolX* and *struc2vec* by 4% and 50% (homogeneity), respectively, providing empirical evidence for our analytical result that GRAPHWAVE is robust to noise in the edge structure. The silhouette scores also show that the clusters recovered by GRAPHWAVE are denser and better separated than for the other methods.

As an example, we show a cycle graph with attached "house" shapes (Figure 3A). We plot 2D PCA projections of GRAPHWAVE's signatures in Figure 3B, confirming that GRAPHWAVE accurately

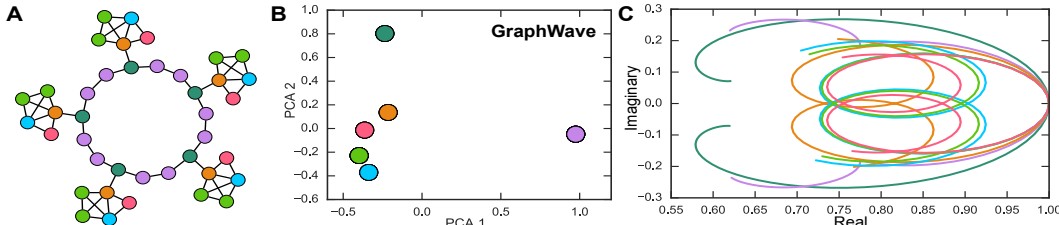

Figure 3: A cycle graph with attached "house" shapes (**A**). 2D PCA projection of GRAPHWAVE's structural signatures. The representation of structurally equivalent nodes overlap, and GRAPHWAVE perfectly recovers the 6 different node types (**B**). Empirical characteristic function for the distribution of the wavelet coefficients (**C**). Color of a node/curve indicates structural role. (Best seen in color.)

| Shapes placed along a cycle graph | | Method | Homogeneity | Completeness | Silhouette |
|---|---|---|---|---|---|
| House | | *RolX* | **1.000** | **1.000** | **1.000** |
| | | *struc2vec* | 0.845 | 0.885 | 0.584 |
| | | GRAPHWAVE | **1.000** | **1.000** | **1.000** |
| House perturbed | | *RolX* | 0.769 | 0.783 | 0.496 |
| | | *struc2vec* | 0.537 | 0.554 | 0.511 |
| | | GRAPHWAVE | **0.802** | **0.807** | **0.566** |
| Varied | | *RolX* | 0.927 | 0.915 | 0.746 |
| | | *struc2vec* | 0.629 | 0.561 | 0.588 |
| | | GRAPHWAVE | **0.943** | **0.948** | **0.867** |
| Varied perturbed | | *RolX* | 0.776 | 0.728 | 0.525 |
| | | *struc2vec* | 0.537 | 0.482 | 0.540 |
| | | GRAPHWAVE | **0.807** | **0.806** | **0.589** |

Table 1: Structural role discovery results for different synthetic graphs. (Best seen in color.) Results averaged over 20 synthetically generated graphs. Dashed lines denote perturbed versions of the basic shapes (obtained by randomly adding and removing edges), node colors indicate structural roles.

distinguishes between nodes with distinct structural roles. We also visualize the resulting characteristic functions (Eq. (2)) in Figure 3C. In general, their interpretation is as follows (Appendix D):

- Nodes located in the periphery of the graph struggle to diffuse the signal over the graph, and thus span wavelets that are characterized by a smaller number of non-zero coefficients. Characteristic functions of such nodes thus span a small loop-like 2D curve.
- Nodes located in the core (dense region) of the graph tend to diffuse the signal farther away and reach farther nodes for the same value of $t$. Characteristic functions of such nodes thus have a farther projection on the $x$ and $y$ axis.

In Figure 3C, different shapes of the characteristic functions capture different structural roles. We note the visual proximity between the roles of the blue, light green and red nodes that these curves carry, as well as their clear difference with the core dark green and purple nodes.

## 5 EXPERIMENTS ON REAL-WORLD GRAPHS

### 5.1 THE ENRON EMAIL GRAPH

**Data and setup.** Nodes represent Enron employees and edges correspond to email communication between the employees (Klimt and Yang, 2004). An employee has one of seven functions in the company (*e.g.*, CEO, president, manager). These functions provide ground-truth information about roles of the corresponding nodes in the network. We use GRAPHWAVE to learn a structural signature for every Enron employee. We then use these signatures to compute the average $\ell_2^2$ distance between every two categories of employees.

**Results.** GRAPHWAVE captures intricate organizational structure of Enron (Figure 4). For example, CEOs and presidents are structurally distant from all other job titles. This indicates their unique position in the email exchange graph, which can be explained by their local graph connectivity patterns standing out from the others. Traders, on the other hand, appear very far from presidents

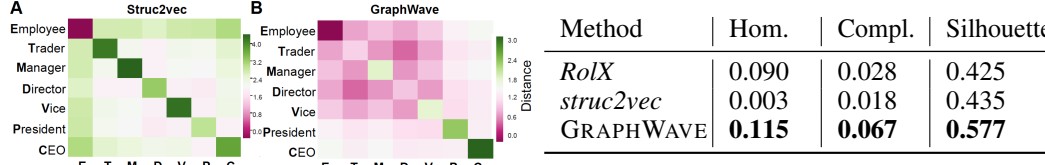

| Method | Hom. | Compl. | Silhouette |
|--------|------|--------|-----------|
| *RolX* | 0.090 | 0.028 | 0.425 |
| *struc2vec* | 0.003 | 0.018 | 0.435 |
| GRAPHWAVE | **0.115** | **0.067** | **0.577** |

Figure 4: Heat maps indicate average distance between roles in the Enron email graph, as determined by *struc2vec* (**A**) and GRAPHWAVE (**B**). Table of comparative statistics for the three algorithms.

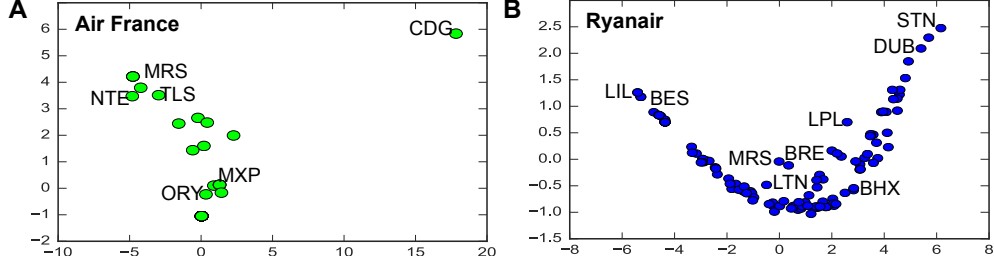

Figure 5: PCA projection of the learned airport structural signatures. **A**: Air France, a major airline, and **B**: Ryanair, a low-cost fare airline. Selected nodes are labeled using three-letter airports codes.

and are closer to directors. In contrast, *struc2vec* is less successful at revealing intricate relationships between the job titles, yielding an almost uniform distribution of distances between every class.

We assess the separation between "top" job titles (CEO and President) and lower levels in the job title hierarchy. GRAPHWAVE achieves 28% better homogeneity and 139% better completeness than *RolX*. We also note that the variability within each cluster of *struc2vec* is higher than the average distance between clusters (dark green colors on the diagonal, and lighter colors on the off-diagonal).

## 5.2 EUROPEAN AIRLINE GRAPHS

**Data and setup.** The airline graphs are taken from the list of airlines operating flights between European airports (Cardillo et al., 2013). Each airline is represented with a graph, where nodes represent airports and links stand for direct flights between two airports. Given an airline graph, we use GRAPHWAVE to learn structural signature of every airport. We then create a visualization using PCA on the learned signatures to layout the graph on a two-dimensional structural space.

**Results.** Figure 5 shows graph visualizations of two very different airlines. Air France is a national French airline, whose graph has the so-called hub and spoke structure, because the airline is designed to provide an almost complete coverage of the airports in France. We note that the signature of CDG (Charles De Gaulle, which is Air France's central airport) clearly stands out in this 2D projection, indicating its unique role in the network. In contrast to Air France, Ryanair is a low-cost airline whose graph avoids the overly centralized structure. Ryanair's network has near-continuous spectrum of structural roles that range from regional French airports (Lille (LIL), Brest Bretagne (BES)) all the way to London Stansted (STN) and Dublin (DUB). These airlines have thus developed according to different structural and commercial constraints, which is clearly reflected in their visualizations.

## 6 CONCLUSION

We have developed a new method for learning structural signatures in graphs. Our approach, GRAPHWAVE, uses spectral graph wavelets to generate a structural embedding for each node, which we accomplish by treating the wavelets as a distributions and evaluating the resulting characteristic functions. Considering the wavelets as distributions instead of vectors is a key insight needed to capture structural similarity in graphs.

Our method provides mathematical guarantees on the optimality of learned structural signatures. Using spectral graph theory, we prove that structurally equivalent/similar nodes have near-identical/similar structural signatures in GRAPHWAVE. Experiments on real and synthetic networks provide empirical evidence for our analytical results and yield large gains in performance over state-of-the-art baselines. For future work, these signatures could be used for transfer learning, leveraging data from a well-explored region of the graph to infer knowledge about less-explored regions.

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

NOTATION

In the appendix, we use the same notation as in the main paper. However, in the main paper, we dropped the explicit dependency of the heat kernel wavelet $\Psi$ on the scaling parameter $s$. We note that we explicitly keep this dependency throughout the appendix as we study the relationship between heat kernel wavelets and the scaling parameter.

## A    KNOWN PROPERTIES OF HEAT KERNEL WAVELETS

We here list five known properties of heat kernel wavelets that we later use to derive a method for automatic selection of the scaling parameter in Appendix C.

P1. The heat kernel wavelet matrix $\Psi^{(s)}$ is **symmetric**:
$$\Psi_{ma}^{(s)} = \delta_m^T U e^{-s\Lambda} U^T \delta_a = \delta_a^T U e^{-s\Lambda} U^T \delta_m = \Psi_{am}^{(s)}.$$

P2. **First eigenvector:** By definition of the graph Laplacian $L = D - A$, the vector $\mathbb{1}$ is an eigenvector of $L$, corresponding to the smallest eigenvalue $\lambda_1 = 0$. This means that:
$$U_1 = 1/\sqrt{N}\mathbb{1} \implies U^T\mathbb{1} = (\sqrt{N}, 0, 0, \ldots, 0)^T = \sqrt{N}\delta_1, \qquad (**)$$
where the last equality follows from the fact that eigenvectors in $U$ are orthogonal.

P3. **Scaling inequality:** Using the Cauchy Schwartz inequality, we get:
$$(\Psi_{ma}^{(s)})^2 = (\sum_{j=1}^N e^{-\lambda_j s} U_{aj} U_{mj})^2 \leq (\sum_{j=1}^N (e^{-\frac{\lambda_j s}{2}} U_{aj})^2)(\sum_{j=1}^N (e^{-\frac{\lambda_j s}{2}} U_{jm})^2) = \Psi_{aa}^{(s)} \Psi_{mm}^{(s)}.$$
It thus follows that: $\Psi_{ma}^{(s)} \leq \max(\Psi_{aa}^{(s)}, \Psi_{mm}^{(s)})$.

P4. **Convergence:** We use the fact that when $s \to \infty$, the value of $e^{-s\lambda}$ converges to 0 for any non-zero eigenvalue $\lambda$ of the graph Laplacian. Combining this fact with the spectral graph wavelet definition in Eq. (1), we get:
$$\forall a, m, \quad \lim_{s\to\infty} \Psi_{am}^{(s)} = \sum_{j=1}^N \lim_{s\to\infty} e^{-s\lambda_j} U_{aj} U_{mj} = U_{a1} U_{m1} = 1/N,$$
since it follows from Eq. (**) that $U_{a1} = U_{1a} = 1/\sqrt{N}$ for every $a$. The diffusion over the network thus converges to a state where all nodes have an identical temperature $1/N$.

P5. **Convergence rate:** To analyze the convergence rate of heat kernel wavelets as a function of $s$, we expand $|\Psi_{aa}^{(s+1)} - \frac{1}{N}|$ using the spectral graph wavelet definition in Eq. (1):
$$|\Psi_{aa}^{(s+1)} - \frac{1}{N}| = |\sum_{j=2}^N e^{-\lambda_j(s+1)} U_{ja}^2| \leq |\sum_{j=2}^N e^{-\lambda_j s} U_{ja}^2| \leq |\Psi_{aa}^{(s)} - \frac{1}{N}|,$$
where:
$$|\Psi_{aa}^{(s)} - \frac{1}{N}| = e^{-\lambda_2 s} |\underbrace{\sum_{j=2}^N e^{-(\lambda_j - \lambda_2)s} U_{ja}^2}_{\text{decreasing function of } s}|.$$

Taking everything together, we get: $e^{-\lambda_2 s} U_{a2}^2 \leq |\Psi_{aa}^{(s)} - \frac{1}{N}| \leq e^{-\lambda_2 s}(1 - \frac{1}{N})$. It thus follows that: $|\Psi_{aa}^{(s+1)} - \frac{1}{N}|/|\Psi_{aa}^{(s)} - \frac{1}{N}| \leq 1$.

## B    PROPERTIES OF HEAT KERNEL WAVELET DISTRIBUTIONS

We prove three propositions about distributions generated by heat kernel wavelets that will be used in Appendix C.

**Proposition 1.** *The mean of heat kernel wavelet $\Psi_a^{(s)}$ is equal to $\mu_a^{(s)} = \frac{1}{N}\sum_{m=1}^N \Psi_{ma}^{(s)} = \frac{1}{N}$. The mean $\mu_a^{(s)}$ is thus independent of the value of scaling parameter $s$ and node $a$.*

*Proof.* We expand the mean of heat kernel wavelet $\Psi_a^{(s)}$ using spectral graph wavelet definition in Eq. (1) and Property 2 (P2) from Appendix A:

$$\mu_a^{(s)} = \frac{1}{N} \sum_{m=1}^{N} \Psi_{ma}^{(s)} = \frac{1}{N} \mathbb{1}^T U e^{-\Lambda s} U \delta_a = \frac{1}{\sqrt{N}} \delta_1^T e^{-\Lambda s} U \delta_a$$

$$= \frac{1}{\sqrt{N}} \delta_1^T (e^{-\lambda_1 s} U_{1a}, e^{-\lambda_2 s} U_{2a}, \ldots, e^{-\lambda_N s} U_{Na})^T = \frac{1}{\sqrt{N}} e^{-\lambda_1 s} U_{1a} = \frac{1}{N}.$$

$\square$

**Proposition 2.** *The heat kernel wavelet coefficient $\Psi_{aa}^{(s)}$ at the initial source node $a$ is a monotonically decreasing function of scaling parameter $s$. Its value is bounded by:* $\frac{1}{N} \leq \Psi_{aa}^{(s)} \leq 1$.

*Proof.* This follows directly using definition $\Psi_{aa}^{(s)} = \sum_{j=1}^{m} e^{-\lambda_j s} U_{ja}^2$ and the scaling inequality in Property 3 (P3). This way, we get:

$$\Psi_{aa}^{(\infty)} = \frac{1}{N} \leq \Psi_{aa}^{(s)} \leq 1 = \Psi_{aa}^{(0)}.$$

$\square$

Additionally, for any $m \neq a$, the wavelet coefficient $\Psi_{ma}^{(s)}$ is non-negative and bounded. Specifically, the wavelet coefficient can be written as: $\Psi_{ma}^{(s)} = \sum_{j=1}^{m} e^{-\lambda_j s} U_{ja} U_{jm} = \frac{1}{N} + \sum_{j=2}^{N} e^{-\lambda_j s} U_{ja} U_{jm}$. It can be bounded by: $0 \leq \Psi_{ma}^{(s)} \leq \max(\Psi_{aa}^{(s)}, \Psi_{mm}^{(s)})$ (Property 4 (P4)).

**Proposition 3.** *The variance of heat kernel wavelet $\mathrm{Var}[\Psi_a^{(s)}]$ is a strictly decreasing function of scaling parameter $s$.*

*Proof.* We use the definition of variance to get:

$$\mathrm{Var}[\Psi_a^{(s)}] = \frac{1}{N} \sum_{m=1}^{N} (\Psi_{ma}^{(s)} - \mu_a^{(s)})^2 = \frac{1}{N} \sum_{m=1}^{N} (\Psi_{ma}^{(s)})^2 - \frac{1}{N^2}.$$

We rewrite the sum on the far right hand-side using the symmetry of wavelet matrix $\Psi$ (Property 1):

$$\sum_{m=1}^{N} (\Psi_{ma}^{(s)})^2 = \sum_{m=1}^{N} \Psi_{am}^{(s)} \Psi_{ma}^{(s)} = (\Psi^2)_{aa} = \delta_a^T U e^{-2s\Lambda} U^T \delta_a = \|e^{-s\Lambda} U^T \delta_a\|_2^2 = \sum_{j=1}^{N} e^{-2s\lambda_j} U_{aj}^2,$$

concluding that variance $\mathrm{Var}[\Psi_a^{(s)}]$ is decreasing, since it is sum of functions, all of which decrease as $s$ gets larger. $\square$

## C    SCALING PARAMETER OF HEAT KERNEL

We here develop a method that automatically finds an appropriate range of values for the scaling parameter $s$ in heat kernel $g_s$, which we use in the multiscale version of GRAPHWAVE (Section 2.2)

We find the appropriate range of values for $s$ by specifying an interval bounded by $s_{\min}$ and $s_{\max}$ through the analysis of variance in heat kernel wavelets. Intuitively, whether or not a given value for $s$ is appropriate for structural signature learning depends on the relationship between the scaling parameter and the temporal aspects of heat equation. In particular, small values of $s$ allow little time for the heat to propagate, yielding diffusion distributions (*i.e.*, heat kernel wavelet distributions) that are trivial in the sense that only a few coefficients have non-zero values and are thus unfit for comparison. For larger values of $s$ the network converges to a state in which all nodes have an identical temperature equal to $1/N$ (Property 4 (P4)), meaning that diffusion distributions are data-independent, hence non-informative.

Next we prove Propositions 4 and 5 to provide new insights into the variance and convergence rate of heat kernel wavelets. We then use these results to select $s_{\min}$ and $s_{\max}$.

**Proposition 4.** *Given the scaling parameter $s$, the variance of off-diagonal coefficients in heat kernel wavelet $\Psi_a$ is proportional to:*

$$Var[\{\Psi_{am}^{(s)}; m \neq a\}] \propto \Delta_a^{(0)}\Delta_a^{(2s)} - (\Delta_a^{(s)})^2,$$

*where $\Delta_a^{(s)} = |\Psi_{aa}^{(s)} - \frac{1}{N}|$ is a monotonically decreasing function of $s$.*

*Proof.* Let us denote the mean of off-diagonal coefficients in wavelet $\Psi_a$ by: $\tilde{\mu}_a^{(s)} = \sum_{m \neq a} \Psi_{ma}^{(s)}/(N-1)$. We use the fact that $\sum_{m \neq a} \Psi_{ma}^{(s)} = 1 - \Psi_{aa}^{(s)}$, along with the definition of the variance, to obtain:

$$Var[\{\Psi_{am}^{(s)}; m \neq a\}] = \frac{1}{N-1}\sum_{m \neq a}(\Psi_{ma}^{(s)} - \tilde{\mu}_a^{(s)})^2 = \frac{1}{N-1}\sum_{m \neq a}(\Psi_{ma}^{(s)})^2 - (\tilde{\mu}_a^{(s)})^2$$

$$= \frac{1}{N-1}(\Psi_{aa}^{(2s)} - (\Psi_{aa}^{(s)})^2) - \frac{1}{(N-1)^2}(1 - \Psi_{aa}^{(s)})^2$$

$$= \frac{N}{(N-1)^2}(\Psi_{aa}^{(2s)}\frac{N-1}{N} - (\Psi_{aa}^{(s)})^2 + \frac{2\Psi_{aa}^{(s)}}{N} - \frac{1}{N})$$

$$= \frac{N}{(N-1)^2}((\Psi_{aa}^{(2s)} - \frac{1}{N})\frac{N-1}{N} - (\Psi_{aa}^{(s)} - \frac{1}{N})^2)$$

$$= \frac{N}{(N-1)^2}(\Delta_a^{(0)}\Delta_a^{(2s)} - (\Delta_a^{(s)})^2).$$

$\square$

Proposition 4 proves that the variance is a function of $\Delta_a^{(s)}$. Therefore, to maximize the variance, we must analyze the behavior of $\Delta_a^{(s)}$. To ensure sufficient variability in the distribution of wavelet coefficients, we need to select a range $[s_{min}, s_{\max}]$ that bounds the $\Delta_a^{(s)}$. Our goal thus becomes establishing that $\Delta_a^{(s)}$ is large enough that the diffusion has had time to spread, while remaining sufficiently small to ensure that the diffusion is far from its converged state.

**Proposition 5.** *The convergence of heat kernel wavelet coefficient $\Psi_{am}^{(s)}$ is bounded by:*

$$e^{-\lambda_N \lceil s \rceil}\Delta_a^{(0)} \leq \Delta_a^{(s)} \leq e^{-\lambda_2 \lfloor s \rfloor}\Delta_a^{(0)}.$$

*Proof.* We use Property 5 from Appendix A and induction over $s$ to complete this proof. For a given $s \geq 0$ we analyze:

$$|\Psi_{aa}^{(s+1)} - \frac{1}{N}| = |\sum_{j=2}^{N} e^{-\lambda_j(s+1)}U_{ja}^2| \leq e^{-\lambda_2}|\Psi_{aa}^{(s)} - \frac{1}{N}|,$$

and conclude that: $e^{-\lambda_N} \leq |\Psi_{aa}^{(s+1)} - \frac{1}{N}|/|\Psi_{aa}^{(s)} - \frac{1}{N}| \leq e^{-\lambda_2}$.

Given any $s \in \mathbb{N}$, we use the induction principle to get:

$$e^{-\lambda_N s}|\Psi_{aa}^{(0)} - \frac{1}{N}| \leq |\Psi_{aa}^{(s)} - \frac{1}{N}| \leq e^{-\lambda_2 s}|\Psi_{aa}^{(0)} - \frac{1}{N}|,$$

which immediately yields the desired bound: $e^{-\lambda_N s}\Delta_a^{(0)} \leq \Delta_a^{(s)} \leq e^{-\lambda_2 s}\Delta_a^{(0)}$. Since $\Delta_a^{(s)}$ is smooth increasing function of $s$, we can take the floor/ceiling of any non-integer $s \geq 0$ and this proposition must hold. $\square$

## C.1 SELECTION OF $s_{\max}$

We select $s_{\max}$ such that wavelet coefficients are localized in the network. To do so, we use Proposition 5 and bound $\Delta_a^{(s)}$ by the graph Laplacian's eigenvalues. When the bulk of the eigenvalues leans towards $\lambda_N$, $\Delta_a^{(s)}$ is closer to $e^{-\lambda_N}$ (*i.e.*, lower bound in Proposition 5). When the bulk of the eigenvalues is closer to $\lambda_2$, $\Delta_a^{(s)}$ will lean towards $e^{-\lambda_2}$ (*i.e.*, upper bound in Proposition 5). In each case, the diffusion is localized if $\Delta_a^{(s)}$ is above a given threshold $\eta < 1$. Indeed, this ensures that

$\Delta_a^{(s)}$ has shrunk to at most $\eta * 100$ % of its initial value at $s = 0$, and yields a bound of the form: $\Delta_a^{(s)}/\Delta_a^{(0)} \geq \eta$. The bound implies that: $e^{-\lambda s} \geq \eta$, or $s \leq -\log(\eta)/\lambda$.

To find a middle ground between the two convergence scenarios, we take $\lambda$ to be the geometric mean of $\lambda_2$ and $\lambda_N$. Indeed, as opposed to the arithmetic mean, the geometric mean maintains an equal weighting across the range $[\lambda_2, \lambda_N]$, and a change of $\epsilon\%$ in $\lambda_2$ has the same effect as a change in $\epsilon\%$ of $\lambda_N$. We thus select $s_{\max}$ as: $s_{\max} = -\log(\eta)/\sqrt{\lambda_2\lambda_N}$.

## C.2 Selection of $s_{\min}$

We select $s_{\min}$ to ensure the adequate diffusion resolution. In particular, we select a minimum value $s_{\min}$ such that each wavelet has sufficient time to spread. That is, $\Delta_a^{(s)}/\Delta_a^{(0)} \leq \gamma$. As in the case of $s_{\max}$ above, we obtain a bound of $s \geq -\log(\gamma)/\lambda$. Hence, we set $s_{\min}$ to: $s_{\min} = -\log(\gamma)/\sqrt{\lambda_2\lambda_N}$.

To cover an appropriate range of scales, we suggest setting $\eta = 0.90$ and $\gamma = 0.99$.

# D Visualization of characteristic functions

Here, we study the properties of characteristic functions in GRAPHWAVE. Our goal is to provide intuition to understand the behavior of these functions and how their resulting 2D parametric curves reflect nodes' local topology (see Figure 3C).

We begin by reviewing the definition of the characteristic function.

**Definition 1.** *The empirical characteristic function (Lukacs, 1970) of wavelet $\Psi_a^{(s)}$ is function $\phi_a^{(s)}$ defined as:*

$$\phi_a^{(s)}(t) = \frac{1}{N}\sum_{m=1}^{N} e^{it\Psi_{ma}^{(s)}}, \quad t \in \mathbb{R}.$$

In the phase plot, a given value of $t$ thus yields the following set of coordinates:

$$\phi_a^{(s)}(t) = \begin{pmatrix} \frac{1}{N}\sum_{m=1}^{N}\cos(t\Psi_{ma}^{(s)}) \\ \frac{1}{N}\sum_{m=1}^{N}\sin(t\Psi_{ma}^{(s)}) \end{pmatrix}.$$

By varying $t$, we get a characeric curve in this 2D plane. Here, we note several properties of this curve:

1. **Value at $t = 0$:** $\forall s, \quad \phi_a^{(s)}(0) = \begin{pmatrix} 1 \\ 0 \end{pmatrix}$, independent of the scale or initial source node.

2. **Behavior for $s = 0$:** Only the coefficient corresponding to the initial source node has non-zero value (*i.e.*, $\Psi_{aa}^{(0)} = 1$, and $\forall m \neq a, \Psi_{ma}^{(0)} = 0$). Hence, $\forall t, \phi_a^{(0)}(t) = \frac{1}{N}e^{it} + \frac{N-1}{N}$. The phase plot of the characteristic curve is a circle of radius $\frac{1}{N}$ with period $2\pi$ centered at $\frac{N-1}{N}$.

3. **Behavior as $s \to \infty$:** From Property 4 in Appendix A, the coefficients all converge to the same limit: $\forall m, a, \lim_{s\to\infty}\Psi_{ma}^{(s)} = \frac{1}{N}$.

   This in turn implies that: $\forall t, \lim_{s\to\infty}\phi_a^{(s)}(t) = e^{i\frac{t}{N}}$. Hence, the curve converges to a circle centered at 0, with radius 1 and period $2\pi N$.

4. **Gradient of the curve:** $\nabla_t\phi_a^{(s)}(t) = \frac{i}{N}\sum_{m=1}^{N}\Psi_{am}^{(s)}e^{it\Psi_{am}^{(s)}}$, or equivalently, in the phase plot:

$$\nabla_t\phi_a^{(s)}(t) = \begin{pmatrix} -\frac{1}{N}\sum_{m=1}^{N}\Psi_{ma}^{(s)}\sin(t\Psi_{ma}^{(s)}) \\ \frac{1}{N}\sum_{m=1}^{N}\Psi_{ma}^{(s)}\cos(t\Psi_{ma}^{(s)}) \end{pmatrix}.$$

   Since all the wavelet coefficients are non-negative, the curve is thus directed counter-clockwise.

