# OpenReview forum: "Spectral Graph Wavelets for Structural Role Similarity in Networks"
_ICLR.cc/2018/Conference — Reject_

### Official Review · AnonReviewer2 · 2017-11-28
**This paper uses spectral graph wavelet diffusion patterns of a node’s local neighborhood to embed the node in a low-dimensional space.  In this low-dimensional space, nodes with similar local neighborhoods are close to each other even if they are far in graph distance.  The methodology of the paper is fine but the experiments section is weak and the paper misses clear connections to sociology.**

**Rating:** 5
**Confidence:** 5

**Review:**

The term "structural equivalence" is used incorrectly in the paper.  From sociology, two nodes with the same position are in an equivalence relation.  An equivalence, Q, is any relation that satisfies these three conditions:
  - Transitivity: (a,b), (b,c) ∈ Q ⇒ (a,c) ∈Q
  - Symmetry: (a, b) ∈ Q if and only if (b, a) ∈Q
  - Reflexivity: (a, a) ∈Q

There are three deterministic equivalences: structural, automorphic, and regular.

From Lorrain & White (1971), two nodes u and v are structurally equivalent if they have the same relationships to all other nodes.  Exact structural equivalence is rare in real-world networks.

From Borgatti, et al. (1992) and Sparrow (1993), two nodes u and v are automorphically equivalent if all the nodes can be relabeled to form an isomorphic graph with the labels of u and v interchanged.

From Everett & Borgatti (1992), two nodes u and v are regularly equivalent if they are equally related to equivalent others.

Parts of this statement are false: "A notable example of such approaches is RolX (Henderson et al., 2012), which aims to recover a soft-clustering of nodes into a predetermined number of K distinct roles using recursive feature extraction (Henderson et al., 2011)."  RolX (as described in KDD 2012 paper) uses MDL to automatically determine the number of roles.

As indicated above, this statement is also false: "We note that RolX requires the number of desired structural classes as input, ...".

The paper does not discuss how the free parameter d (which represents the number of evenly spaced sample points)  is chosen.

This statement is misleading: "In particular, a small perturbation of the graph yields small perturbations of the eigenvalues."  What is considered a small perturbation?  One can delete an edge (seemingly a small perturbation) and change the eigenvalues of the Laplacian dramatically --  e.g., deleting an edge that increases the number of connected components.

The barbell graph experiment seemed contrived.  Why would one except such a graph to have 8 classes?  Why not 3?  One for cliques, one for the chain, and one for connectors of the clique to the chain.

In Section 4.2, how many roles were selected for RolX?

The paper states: "Furthermore, nodes from different graphs can be embedded into the same space and their structural roles can be compared across different graphs."  Experiments were not conducted to see how the competing approaches such as RolX compare with GraphWave on transfer learning tasks.

Gilpin et al (KDD 2013) extended RolX to incorporate sparsity and diversity constraints on the role space and showed that their approach is superior to RolX on measuring distances.  This is applicable to experiments in Figure 4.

I strongly recommend running experiments that test the predictive power of the roles found by GraphWave.

---

> ### Author Response · Authors · 2017-12-12
> **Response to the reviewer's comments: clarifications, additional references and predictive power**
>
>
> We thank the reviewer for the detailed comments and questions regarding our submission. Here, we try to clarify some details and address the reviewer’s concerns:
>
> #1: The term "structural equivalence" is used incorrectly in the paper.
>
> We emphasize that we are using the same definition of structural equivalence as Lorrain and White, 1971. However, perfect structural equivalence, as the reviewer points out, is extremely rare in real-world networks. Therefore, instead of looking for nodes with exact equivalence, we instead recover a low-dimensional embedding, or a structural signature, to find structurally similar nodes. We note that this notion of structural similarity is a commonly-used term in network science (Airoldi et al., 2008; Hoff et al., 2008; Newman, 2011; Henderson et al., 2012; Grover et al., 2016; Ribeiro et at., 2017;  etc).
>
> #2: RolX uses MDL to automatically determine the number of roles.
>
> While RolX algorithm requires a pre-determined number of clusters, the reviewer is correct in mentioning that the RolX authors do include a method of automatically selecting this number using MDL. We thank the reviewer for this comment and will update the manuscript to reflect this fact. We note however that in our experiments, as we point out in Section 4 of our paper, we used RolX as an oracle estimator (providing it with the “correct” number of classes, the best-case scenario for RolX).
>
> #3: The paper does not discuss how the parameter d is chosen.
>
> We set d=100 in all experiments. This parameter corresponds to the number of sampling points along the characteristic parametric curves (example is shown in Figure 3C). We have not put any special effort to tune this parameter.
>
> #4: What is considered a small perturbation?  One can delete an edge (seemingly a small perturbation) and change the eigenvalues of the Laplacian dramatically.
>
> As correctly highlighted by the reviewer, a small perturbation cannot be defined simply through the Hamming distance between the original and perturbed adjacency matrices. In our paper, we use definition of a small perturbation as defined in Spielman, Spectral Graph Theory (Chapter 16), 2011. That is, a small perturbation of the k-hop neighborhood corresponds to a set of edge additions/deletions that have a small impact on the graph Laplacian L. As proved by Spielman and studied by Milanese et al., 2010, in this setting, the perturbation induced on the eigenspectrum and the eigenvectors of L is small. Thus, the difference between the original Laplacian L and the perturbed Laplacian \tilde{L} is small as well (sup ||L^k -\tilde{L}^k|| <eps). We thank the reviewer for pointing out that this definition was unclear, and we will make sure to clarify it in the revised paper.
>
> #5: Why would one expect the barbell graph to have 8 classes?  Why not 3?  One for cliques, one for the chain, and one for connectors of the clique to the chain.
>
> Using the definition of structural equivalence as defined by Lorrain and White, 1970, the barbell graph has exactly 8 structurally equivalent classes (one corresponding to the nodes in the cliques, and the seven others comprising the nodes in the chain at a given distance level to the cliques). We note that GraphWave can recover all 8 classes, whereas RolX is only able to discover 3 classes, indicating that GraphWave can recover fine grain structural information.
>
> #6: In Section 4.2, how many roles were selected for RolX?
>
> Please see our answer to Comment #2 above. We used RolX as an oracle estimator, providing it with the correct number of classes, the best-case scenario for RolX.
>
> #7: Experiments were not conducted on transfer learning tasks.
>
> While we mention transfer learning as a potential application of this work, we had originally left the formal analysis of such methods to future work, as we discussed in the conclusion. However, due to the reviewer’s comments we ran a new transfer learning experiment (see the APPENDIX in our response to Reviewer 1). These results show that GraphWave outperforms several state of the art methods for the transfer learning task.
>
> #8: We did not compare with Gilpin et al., 2013.
>
> We thank the reviewer for pointing out the reference to Gilpin et al., 2013, and their method, GLRD. We were not aware of it and will add it to the related work. While the code for the method was not published online by the authors, we implemented a simplified version of their method ourselves (incorporating sparsity constraints). In additional experiments described in our response to Reviewer 1 (APPENDIX), GraphWave outperformed GLRD by 260% in homogeneity, 430% in completeness, and by 500% in silhouette score.

---

### Official Review · AnonReviewer1 · 2017-12-04
**The method is based on a principled derivation, but have not compared with other state-of-the-art**

**Rating:** 5
**Confidence:** 4

**Review:**

The paper derived a way to compare nodes in graph based on wavelet analysis of graph laplacian. The method is correct but it is not clear whether the method can match the performance of state-of-the-art methods such as graph convolution neural network of Duvenaud  et al. and Structure2Vec of Dai et al. in large scale datasets.
1. Convolutional Networks on Graphs for Learning Molecular Fingerprints. D Duvenaud et al., NIPS 2015.
2. Discriminative embeddings of latent variable models for structured data. Dai et al. ICML 2016.

---

> ### Author Response · Authors · 2017-12-12
> **Response to the reviewer's comments: complementary state-of-the art**
>
> We thank the reviewer for pointers to these papers, which we have carefully reviewed. However, we would like to explicitly point out that the methods developed in the mentioned papers have a different goal to GraphWave’s. In particular, both Molecular Fingerprints and Structure2Vec are solving a  “graph-level embedding” problem, which converts an entire graph into a single low-dimensional vector. In contrast, GraphWave is solving the “node-level embedding” problem, where it generates a low-dimensional vector for each node based on node’s structural role in the graph -- which is why we had not originally intended comparing GraphWave to these methods.
>
> However, in certain settings, both Molecular Fingerprints and Structure2Vec can be compared with GraphWave. Specifically, these two methods yield node embeddings as a by-product of their algorithm, but only in supervised settings across multiple graphs where the graphs have “ground truth” labels. Following the reviewer’s suggestion, we developed an additional experiment to compare GraphWave with these methods in a specific supervised setting (see APPENDIX below). We note that GraphWave is much more general, and can yield node embeddings on a single graph, or across multiple unlabeled graphs, something that Molecular Fingerprint and Structure2Vec are unable to do. As shown in the experiments (APPENDIX, results), GraphWave outperformed Molecular Fingerprints by 37% in homogeneity score, 11% in completeness, and 890% in silhouette score. Additionally, GraphWave outperformed Structure2Vec by 4% in homogeneity, though Structure2Vec had a 7% higher completeness and 54% higher silhouette score.
>
> Overall, GraphWave outperforms the state-of-the-art in unsupervised settings (see the experiments in the paper) and yields very strong performance in supervised settings, even when compared against supervised methods (as shown in the APPENDIX here).
>
> To address these comments by the reviewer, we will add both references to the related work section.
>
> ------------------------------------------------
> APPENDIX: Additional experiments.
>
> *Experimental setup.*
> The goal of these experiments is to assess the predictive power of embeddings. That is, we analyze how well we can recognize structural similarity of nodes across different graphs. Note that the setup is a slight adaptation of the experiments in the paper. This was required in order to work across multiple graphs --- which was necessary to evaluate Molecular Fingerprints and Structure2Vec methods --- rather than within a single graph.
>
> In particular, we generate 200 graphs, with ground truth labels corresponding to the true structural classes of each node. Each graph was generated as follows:
> We generate its basis (a cycle, as in Figure 3A) of different (random) length.
> We plant a random number of different shapes (houses, fans or stars, as shown in Figure 3A) on this cycle. Our experiment is set up so that with 60% probability, the graph only comprises one type of shape repeated multiple times (20% house, 20% fan, 20% stars), and with 40% chance, the graph comprises all of these shapes in varied numbers.
> We have fixed a priori the scale in GraphWave to s=3. We trained Neural Fingerprints and Structure2Vec by providing each graph with a label (1: house, 2: fan, 3: star, 4: varied). We note that in this setting, the graph labels highly correlated with the structural roles of the nodes. This gives the supervised methods (Molecular Fingerprints and Structure2Vec) an advantage over the unsupervised GraphWave approach. This is necessary because without these labels, the supervised methods cannot be applied.
>
> We run each algorithm, then fit k-means on the embeddings of the first 150 graphs to try to recover the 15 different structural roles of this experiments. We evaluate the performance of the clustering on the remaining 50 graphs in the test set.
>
> *Results.*
> Results are shown in the following table.
>
> Method					                             | Homogeneity | Completeness | Silhouette
> -----------------------------------------------------------------------------------------------------------------------------
> RolX (Henderson et al., 2012)			                  0.688		     0.352		       0.466
> GLRD (Gilpin et al., 2013)				                  0.329		     0.175		       0.101
> Structure2Vec (Dai et al., 2016)			                  0.825		     0.811		       0.890
> Molecular Fingerprints (Duvenaud et al., 2015)      0.626	             0.681		       0.065
> GraphWave (this paper)				                  0.860		     0.756		       0.579

---

### Official Review · AnonReviewer4 · 2017-12-12
**The method is ad-hoc, there is no genuine experimental baseline, the mathematical derivations make the paper look more sophisticated but do not provide actual insight.**

**Rating:** 3
**Confidence:** 4

**Review:**

The paper proposes a method for quantifying the similarity between the local neighborhoods of nodes in a graph/network.

There are many ways in which such a distance/similarity metric between nodes could be defined. For example, once could look at the induced subgraph G_i formed by the k-neighborhood of node i, and the induced subgraph G_j of the k-neighborhood of node j, and define the similarity as k(G_i,G_j) where k is any established graph kernel. Moreover, the task is unsupervised, which makes it hard to compare the performance of different methods. Most of the experiments in the paper seem a bit contrived.

Regarding the algorithm, the question is: “sure, but why this way?”. The authors take the heat kernel matrix on the graph, treat each column as a probability distribution, compute its characteristic function, and define a distance between characteristics functions. This seems pretty arbitrary and heuristic. I also find it confusing that they refer to the heat kernel as wavelets. The spectral graph wavelets of Hammond et al is a beautiful construction, but, as far as I remember, it is explicitly emphasized that the wavelet generating function g must be continuous and satisfy g(0)=0. By setting g(\lambda)=e^{-s \lambda}, the authors just recover the diffusion/heat kernel of the graph. That’s not a wavelet. Why call this a “spectral graph wavelet” approach then? The heat kernel is much simpler. I find this misleading.

I also feel that the mathematical results in the paper have little depth. Diffusion is an inherently local process. It is natural then that the diffusion matrix can be approximated by a polynomial in the Laplacian (in fact, it is sufficient to look at the power series of the matrix exponential). It is not surprising that the diffusion function captures some local properties of the graph (there are papers by Reid Andersen/ Fan Chung/ Kevin Lang, as well as by Mahoney, I believe on localized PCA in graphs following similar ideas). Again, there are many ways that this could be done. The particular way it is done in the paper is heuristic and not supported by either math or strong experiments.

---

> ### Author Response · Authors · 2017-12-14
> **RE: Response to the reviewer's comments**
>
> We thank the reviewer for his or her comments. We address them below.
>
> #1: “There are many ways in which such a distance could be defined [...].”
>
> While we agree with the reviewer that there are many sensible definitions of node similarity, we would like to note that our goal here was broader than plain similarity search. In particular, we aimed to define a structural signature, i.e., an embedding for each node, which requires O(N) memory, instead of O(N^2) for the kernel-based pairwise comparisons suggested by the reviewer. We note that learning embeddings for graphs is a very common problem in machine learning (Henderson et al., 2012; Grover et al., 2016; Ribeiro et al., 2017)
>
> #2: "Moreover, the task is unsupervised, which makes it hard to compare the performance of different methods. Most of the experiments in the paper seem a bit contrived."
>
> We respectfully disagree with this characterization of our unsupervised task. Specifically, we developed multiple synthetic experiments and two real-world case studies to quantitatively compare GraphWave with two state-of-the-art approaches for solving the same unsupervised problem (struc2vec and RolX). Our experiments built upon those from these recent papers (e.g., the Barbell graph was a direct adaptation of an experiment in Ribeiro et al., 2017). In addition, we developed experiments that evaluated GraphWave in a variety of more complex settings (see Sections 4.1 and 4.2 in the paper). Overall, we believe that these experiments sufficiently demonstrate the benefits of GraphWave. However, we would appreciate any additional feedback on specific examples of how to further improve our experiments.
>
> #3: Concerns about spectral graph wavelet transform (SGWT) definition.
>
> The reviewer is correct in stating that the SGWT requires g(0)=0. However in Section 4.2 of their paper, Hammond et al., 2010 introduces a "second class of waveforms," which they call "spectral graph scaling functions." As Hammond et al., 2010 states, these waveforms are "analogous to the lowpass residual scaling functions from classical wavelet analysis.[...] They will be determined by a single real valued function h : R+ → R, which acts as a lowpass filter, and satisfies h(0) > 0 and h(x) → 0 as x → ∞. " As we mention in Section 2.1, the heat kernel in GraphWave is a function of this class, and as such, it falls under Hammond et al.’s general SGWT framework. Because our work directly builds on Hammond et al.’s definition, we use the term spectral graph wavelet, rather than “heat kernel”, even though either term would be appropriate.
>
> #4: ''sure but why this way?'' [...] Diffusion is a local process [...] There are many ways in which this could be done".
>
> We agree with the reviewer in that GraphWave relies on an inherently local diffusion process. However, comparing diffusions across nodes in the graph to recover structural similarities is a tricky problem. Without an a-priori-known one-to-one mapping between neighborhoods, we are not aware of a computationally tractable method for comparing diffusions localized in different parts of the graph. For this reason, we suggested considering these diffusions as distributions, thus making the signature permutation-invariant to the labeling of the nodes.
>
> We thank the reviewer for taking the time to read our response, and we hope that he or she will consider our arguments and help us improve our methodology.

---

### Decision · Program_Chairs · 2018-01-29
**ICLR 2018 Conference Acceptance Decision**

**Decision:**

Reject

**Comment:**

The reviewers present strong concerns about the lack of novelty in the paper. Further there are strong concerns about how the experiments are conducted. I recommend the authors to carefully go through the reviews.